# Deformation Field Analysis of Small-Scale Model Experiment on Overtopping Failure of Embankment Dams

**Qiang Lu, Yanchang Gu \*, Shijun Wang, Xiandong Liu and Hong Wang**

Department of Dam Safety and Management, Nanjing Hydraulic Research Institute, No. 223 Guangzhou Road, Nanjing 210098, China; lulus@hhu.edu.cn (Q.L.)
\* Correspondence: ycgu@nhri.cn

**Abstract:** There are a large number of reservoir dams in China, of which embankment dams account for more than 90%, and public safety will be seriously endangered in case of dam failure. Overtopping is the leading cause of dam failure, and the existing research mainly focuses on the study of the failure process, with less research on the change in the deformation field during the failure process. In this study, the measured deformation field data of a modeled embankment dam during the whole process of impoundment, operation, and failure were obtained by carrying out indoor small-scale model experiments of overtopping failure, embedding inclinometers inside the dam body, and setting vertical displacement measurement markers on the surface. A refined analysis of the measured deformation data shows that the dam body displaces vertically downward during the impoundment stage and the vertical displacement at the dam crest has the largest amplitude; the internal horizontal displacement changes to the left bank and downstream side, and the amplitude of the internal horizontal displacement (upstream and downstream direction and dam axis direction) on the right dam sections is more significant than that in the middle of the dam; during the breaching stage, the time sequence of the sudden change in each internal horizontal displacement measuring point is from the downstream side to the upstream side and from the higher elevation to the lower elevation, which is basically consistent with the process of overtopping of embankment dams; and the overall sudden change in left and right bank horizontal displacements within the downstream side of the dam crest and the downstream side of the dam body gauges is significant, and the sudden change in upstream and downstream horizontal displacement (U&D HD) within the downstream side of the dam crest gauges is significant. The experimental analysis results can support the disaster mechanism of embankment dam failure and the theory of early warning of failure.

**Keywords:** embankment dam; overtopping failure; model experiments; analysis of the deformation field

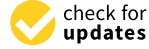



## 1. Introduction

In order to make rational use of water resources, a large number of dams have been constructed in major river basins around the world, playing an essential role in irrigation, power generation, water supply, navigation, and defense against flooding as an essential water-retaining structure [1]. Embankment dams are widely used due to their easy availability of materials, structure simplicity, and construction ease [2]. According to official statistics, as of 2020, China has built 98,566 dams with a total capacity of 930.6 billion $m^3$, and the number of embankment dams accounts for more than 90% of the total number of dams built [3]. With the current global warming accelerated in evolution and extreme climate events characterized by frequent, widespread, and heavy occurrences [4], there is also a risk of dam failure in reservoirs. Once a dam breaches, it will cause considerable losses to the lives and properties of the people downstream. For example, in 1975, the "75.8" flood in Henan Province, China, led to the overtopping and breaching of the Banqiao and Shimantan embankment dams, and the maximum instantaneous flow rate of the breaching flood was 79,000 $m^3/s$; the flood water injected into the downstream was 70.1 million $m^3$

within six hours, which directly led to an affected population of 5.4 million and the deaths of 26,000 people [5,6].

Due to the high proportion of embankment dams and the low flood-control standard of early reservoirs, 94.1% of the 3541 dams that failed in China from 1954 to 2020 were embankment dams, according to the Dam Safety Management Center of the Ministry of Water Resources [7]. According to the statistics of previous dam failure cases at the domestic and international levels, over 50% of dam failure accidents are caused by overtopping [7,8]. The overtopping of reservoirs is usually caused by reservoirs experiencing overtopping floods, i.e., insufficient reservoir discharge capacity or improper reservoir operation and management [8]. In the face of the frequent occurrence of extreme weather and the complexity of the dam operating environment, further research on the overtopping failure process and failure mechanism of embankment dams is of great significance for disaster prevention and mitigation, as well as for the protection of people's lives and properties downstream.

Physical modeling of dam failure is an essential means of reproducing the process of dam failure and revealing the mechanisms of dam failure. Since the dam failure process involves many disciplines, such as hydraulics, soil mechanics, and material science, it is not easy to meet all similar requirements in the model design, and the experiment results will inevitably have deviations, which is the influence of the scale effect. In order to reduce the influence of the scale effect, many scholars carried out a large number of model tests with different scales as well as on-site large-scale model tests. They concluded that the main similarity criteria that should be satisfied for the model test of overtopping failure of embankment dams are geometric similarity, kinematic similarity, dynamic similarity, gravity similarity, and similarity of the development process of the breach [9,10].

In the early tests, similarity to the prototype dam materials was achieved by using dam materials, particle gradation, and construction quality similar to that of the prototype dam or by using a series of modeling tests to gradually approximate the methods to achieve similarity to the prototype materials. With the deepening of understanding, scholars introduced the sediment transport theory in the study of dam failure [11,12], which argues that dam construction materials should meet the similarity of sediment transport and need to ensure that the sand particle Reynolds number is greater than 70 and the water flow Reynolds number is more significant than 1500 during the test to reduce the influence of viscous forces on the test results. Coleman [11] selected the same sediment as the dam prototype and established a similarity criterion for dam failure modeling tests of homogeneous dams with non-cohesive soils based on the similarity principle of sediment movement; Dupont et al. [12] selected the same drag coefficient as the dam prototype and proposed a similarity theory for dam failure modeling tests of homogeneous dams with non-cohesive soils based on the similarity theory of Yalin sediment movement. The similarity criterion they gave was obtained from a series of dam failure tests. Due to the differences between the model and the prototype and the existence of the scale effect, the selection of the similarity ratio of the parameters of the similarity theory cannot wholly rely on the results derived from the similarity theory but also needs to be combined with the purpose of the test and the characteristics of the model test similarity criterion to propose a fit.

The large-scale model test is more similar to the dam failure process, which more accurately reflects the dam failure and flooding processes. Many diverse-scale physical modeling experiments have been conducted in various countries since the 1990s, among which the most representative ones are the IMPACT project supported by the European Union in 2004, which carried out five groups of large-scale field failure model tests with dam heights of 4–6 m [13,14]; the United States Department of Agriculture's Agricultural Research Center (USDA-ARS), which carried out ten groups of field failure tests with dam heights of 1.3–2.3 m [15,16]; and the Nanjing Hydraulic Research Institute (NHRI), which conducted an overtopping failure test on a homogeneous clayey embankment dam with a dam height of 9.7 m [9]. With the above results of field dam failure modeling tests, the primary process and mechanism of embankment dam failure have been more deeply understood. Zhang et al. summarized the results of their overtopping failure tests and

found that the fundamental processes of embankment dam failure are "headcut" backward erosion, "two-helix flow" erosion of the dam crest, and collapse of breach sidewalls [9].

The large-scale breach model test has high input costs and poor test risk controllability. Since the large-scale breaching field test provides a reference for indoor small-scale flume test research design and the small-scale model test is highly fault-tolerant, small-scale model tests can be repeated, thus reducing the influence of uncertainties in the test process on the test phenomena and results. The U.S. National Dam Safety Program (NDSP) [17], the European Union-supported CADAM project [18], and the IMPACT project [19] have conducted several small-scale embankment-dam overtopping-failure model tests to investigate the effects of different particle gradation, dam construction materials, compaction degree, water content, and other factors on the process of overtopping failure in homogeneous embankment dams. Dhiman et al. [20] investigated the effect of changes in water content and compaction effort on the headward erosion and breach flow processes and found that the development of cracks provided channels for water to intrude into the interior of the dam body, which further exacerbated the process of dam failure. Without considering the effect of dam cracks, Amaral et al. [21] compared the dam breaching time and peak breaching flow with different compaction and water content and found that increasing the compaction of the dam body can effectively improve the anti-erosion ability and reduce the erosion resistance of dams below the optimal water content. Verma et al. [22] investigated the effects of crack shape and water content on the breaching process and found that the water content directly affects the crack width, and a decrease in water content leads to faster dam failure time. Dhiman [23] and Kouzehgar [24] studied the effect of dam material on the flow of the dam breach and the development of the breach, and the results showed that a decrease in the average particle size of the dam material accelerates the rate of erosion of the dam; an increase in the number of fine particles in the material increases the failure time to a specific threshold limit and then decreases.

From the above, in the existing large-scale and small-scale physical modeling tests of overtopping failure of embankment dams, the main results obtained focus on the shape of the breach and its development process, the evolution process of the breaching flood, and the influencing factors of the breaching, but there are fewer studies on the process of the deformation field of the dam during the process of encountering the overtopping of embankment dams and the breaching of the dam body.

In this study, an indoor small-scale model experiment of overtopping failure of homogeneous embankment dams was carried out. Through the vertical displacement measurement markers set on the surface of the dam body, the surface deformation process of the dam body before overtopping was obtained. The inclinometers was embedded in the interior of the model and an automatic acquisition device was used to obtain the measured data of the internal deformation field of the modeled embankment dam during the whole process of impoundment, operation, and failure. Through the analysis of the whole process and deformation field data of the model embankment dam, the deformation warning signs of the embankment dam can be identified, which can provide theoretical support for the prediction and early warning of the overtopping failure of the embankment dam.

## 2. Experimental Setup and Test Procedure

### 2.1. Model Similarity Problems

The overtopping failure of embankment dams is a complex water–soil coupling process that is related not only to the water flow conditions but also to the dam materials. The hydrodynamic similarity of the dam failure model is usually according to Froude's similarity ($F_r = v/(gL)^{0.5}$, $\lambda_{Fr} = 1$) [25]. The relationships between the velocity scale $\lambda_u$, the time scale $\lambda t$, the discharge scale $\lambda_Q$, and the length scale $\lambda_l$ are $\lambda_u = \lambda_l^{0.5}$, $\lambda_t = \lambda_l/\lambda_u = \lambda_l^{0.5}$, and $\lambda_Q = \lambda_u\lambda_l^2 = \lambda_l^{2.5}$, respectively. The difficulty of modeling a dam-break experiment is how to keep the dam failure process similar under the premise of ensuring the similarity of flow dynamics [10]. Therefore, it is almost impossible to make all parameters precisely similar to the prototype; only partial similarity can be taken, i.e., an approximate model test.

The ultimate goal of this study is to characterize and quantify the overtopping breaching process. This is not built on a specific (prototype) dam, so it takes the form of approximate model tests [21].

### 2.2. Model Design

The model experiment of overtopping failure of homogeneous embankment dams was carried out in a glass flume at the Nanjing Hydraulic Research Institute experiment research base, and the glass flume is mainly based on the water inlet section, the working section, the water outlet section, and other components. The size of the water inlet section is 200 cm × 200 cm × 200 cm ($L \times W \times H$, the same as below); the working section is made of reinforced glass, with a size of 1000 cm × 100 cm × 120 cm; the water outlet section is equipped with a sedimentation reservoir and a plug-in tailgate, with a size of 150 cm × 100 cm × 120 cm. The experimental embankment dam is a homogeneous earth–rock dam with a height of 50 cm, a width of 20 cm at the dam crest, a length of 100 cm, and upstream and downstream dam slopes of 1:2. To facilitate the embedding of sensors inside the dam, the initial breach was preset at the mid-dam location [26], and the dimensions of the breach and the schematic diagram of the model are shown in Figure 1.

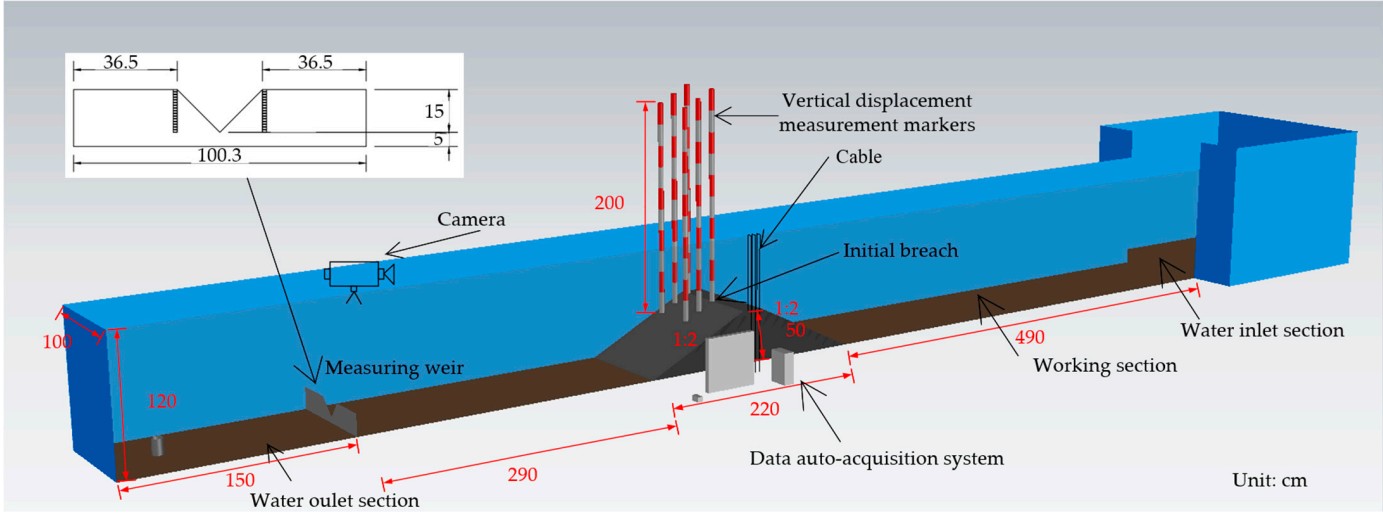

(a)

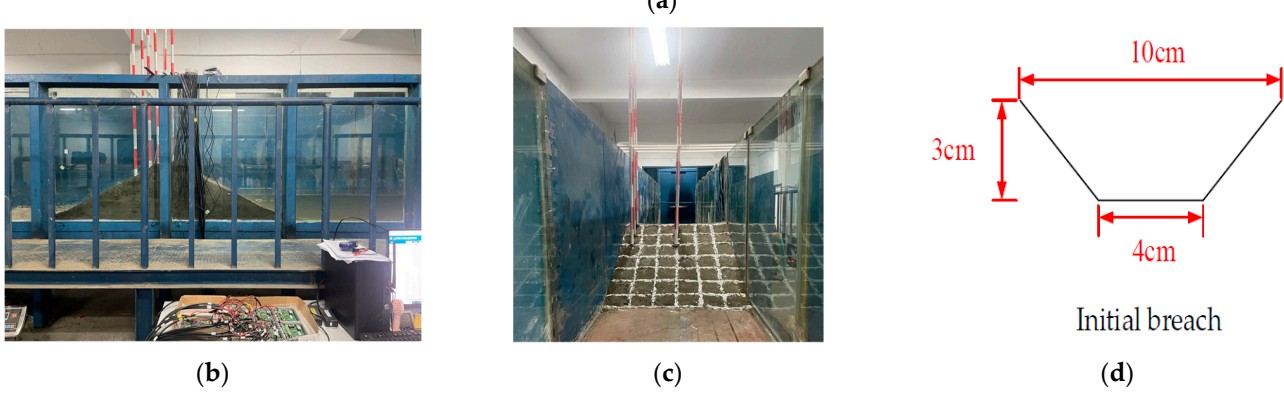

(b)        (c)        (d)

**Figure 1.** (**a**) Schematic diagram of the experimental model; (**b**) side view of the model; (**c**) downstream slope of the dam; (**d**) dimensions of the initial breach.

### 2.3. Dam Materials and Dam Filling Process

Although this experimental study is not specific to a particular prototype dam, the soil materials used in the model need to meet certain conditions to ensure that the results of the overtopping experiment are comparable to those of a homogeneous embankment dam model of similar dimensions. For example, it is recommended that the content of fine-grained soils be between 12% and 50% to ensure that the dams have structural properties with a certain degree of cohesion and that the relative density of the dams be controlled to be between 80% and 90% to ensure the embankment dam's bulk erodibility [21].

The model fill soil material is fine-grained sand (SF); some soil samples were taken and tested in the laboratory, and the soil parameters are shown in Table 1.

**Table 1.** Soil parameters of fill material used for embankment models.

| Soil Classification | Fine-Grained Soil Content (%) | Median Size (mm) | Maximum Dry Density (g/cm$^3$) | Minimum Dry Density (g/cm$^3$) | Hydraulic Conductivity (cm/s) | Water Content (%) | Cohesion (kPa) | Internal Friction Angle (°) |
|---|---|---|---|---|---|---|---|---|
| SF (fine-grained sand) | 10.4 | 0.176 | 1.59 | 1.21 | $6.63 \times 10^{-4}$ | 15 | 14 | 34.1 |

During the experiment, since it was impossible to carry out on-site compaction tests, the dry density of the fill was used to control the compaction of the model, i.e., the relative density of the model was determined first. Then, the relative density formula was used to calculate the dry density achieved during the fill. The dry density was calculated using the soil mass needed to fill the model. The relative density formula is as follows:

$$Dr = \rho_{max}(\rho_d - \rho_{min})/\rho_d(\rho_{max} - \rho_{min}) \tag{1}$$

where $Dr$ is the relative density, $\rho_d$ is the dry density, $\rho_{max}$ is the maximum dry density, and $\rho_{min}$ is the minimum dry density.

According to the dam construction materials and filling requirements proposed in the *Design Code for Rolled Earth–Rock Dams (SL 274-2020)* [27], the hydraulic conductivity of homogeneous dams should not be greater than $1 \times 10^{-4}$ cm/s, and the relative density of sand fillings should not be lower than 0.7. Due to the significant hydraulic conductivity of the modeled soils, in order to avoid seepage failure of the dams before overtopping, the relative density is taken as 0.9. The maximum dry density of the model soil sample is 1.59 g/cm$^3$, and the minimum dry density is 1.21 g/cm$^3$. According to Equation (1), it can be concluded that the dry density to be achieved for the filling is 1.54 g/cm$^3$, and the volume of the model is 0.6 m$^3$, so about 925 kg of sandy soil is required.

The flume sidewall faces immediately adjacent to the dam's sides were waterproofed with Vaseline prior to dam filling to attenuate the bounding effect of the flume sidewalls on the modeled dam. Vaseline not only reduces the confinement of the flume to the modeled dam but also prevents contact leakage at the interface. In order to ensure the homogeneity of the dam body and the reproducibility of the experiment, the dam body was filled in five layers, each with a thickness of 10 cm, and the mass of the required soil material was calculated based on the volume of each layer area. For more visualization of the dam failure process, the model dam was constructed by dividing a grid of approximately 17 cm × 17 cm on the downstream slope of the dam. Figure 2 shows the filling process of the model dam.

Dam filling and instrumentation embedding began at 09:00 on 28 June 2023, and ended at 18:00 on 28 June 2023, lasting nine hours.

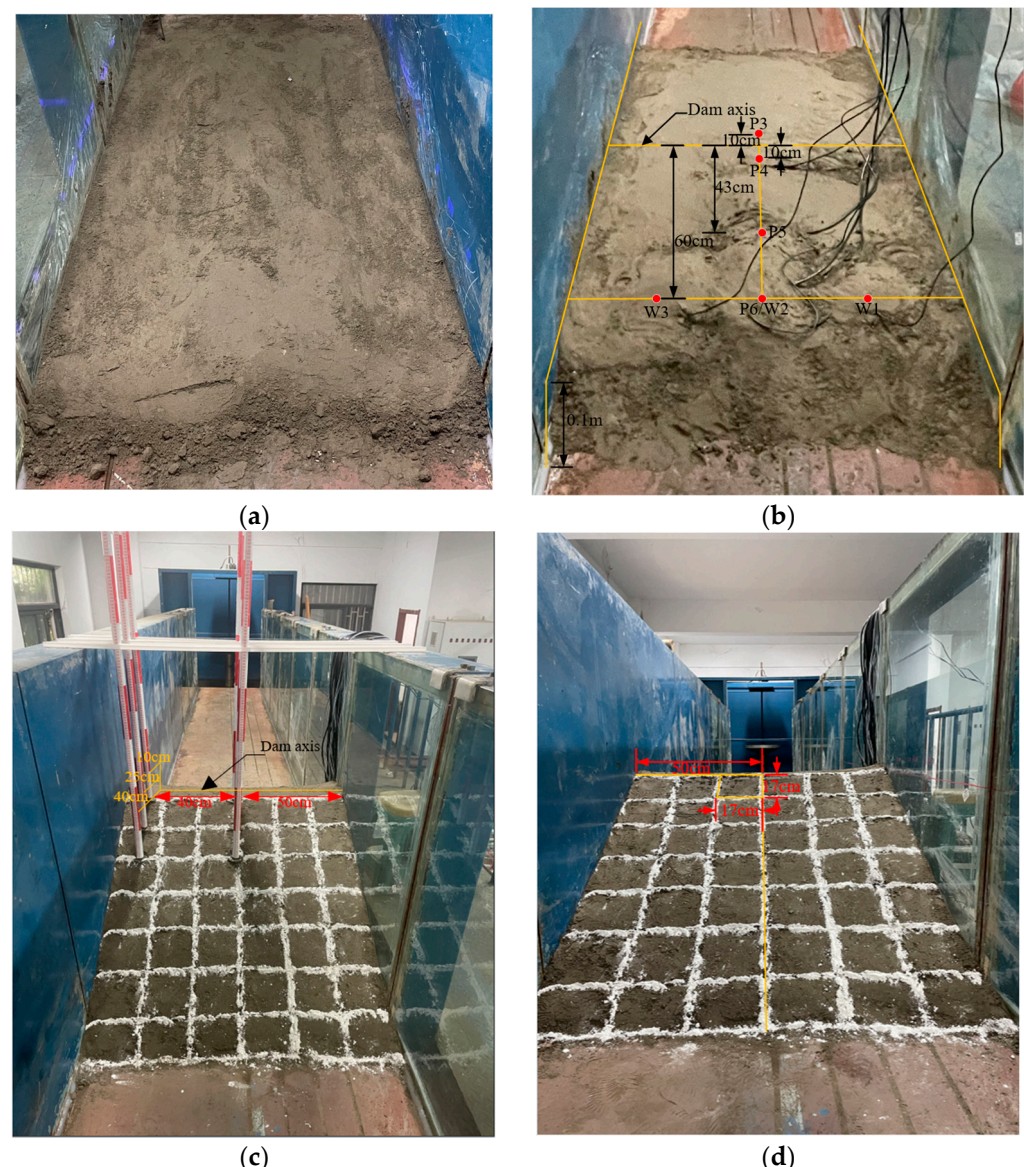

**Figure 2.** Filling process of the model dam: (**a**) dam compaction; (**b**) embedded sensors; (**c**) measurement benchmark installation; (**d**) division grids.

*2.4. Measurement Items and Techniques*

2.4.1. Measurement Items

The measurement program includes environmental variables and deformation, and the specific measurement point layout is detailed in Figures 3 and 4.

(1) Environmental variables: The environmental variables monitored include upstream and downstream water levels and air temperatures. Upstream and downstream water levels are monitored dynamically in real time through water level gauges in the water inlet section of the flume, in conjunction with manual calibration of the water scale.

(2) Deformation: Deformation monitoring includes surface deformation and internal deformation monitoring. The positive and negative signs of displacement changes are specified as follows: vertical displacement downward is positive, horizontal displacement to the left bank and downstream is positive, and vice versa is negative.

(3) Surface deformation: Two surface deformation monitoring sections are set up in the dam body, respectively, in the center section of the dam (section A) and the corresponding position 40 cm to the right of the center section (section B). Three measurement points are arranged in each section, and a measurement rod is used to

measure the vertical displacement of the measurement points [28,29]. The measuring rod is divided into red and white bars (or marking scales). Due to the deformation of the surface of the dam body, and under the action of self-weight, readings are taken through the bar (scale) on the rod in order to measure the displacement.

(4) Internal deformation: Two internal deformation monitoring sections are set up in the dam body, respectively, in the center section of the dam (section A) and the corresponding position 40 cm to the right of the center section (section B). Among them, in section A, one inclinometer is set at each elevation of 35 cm and 20 cm above the dam foundation at axis distances of −10 cm, 10 cm, and 25 cm, respectively, and one inclinometer is set at an elevation of 20 cm above the dam foundation at an axis distance of 40 cm; in section B, one inclinometer is set at each elevation of 35 cm and 20 cm above the dam foundation at an axis distance of 10 cm. A total of nine inclinometers are used.

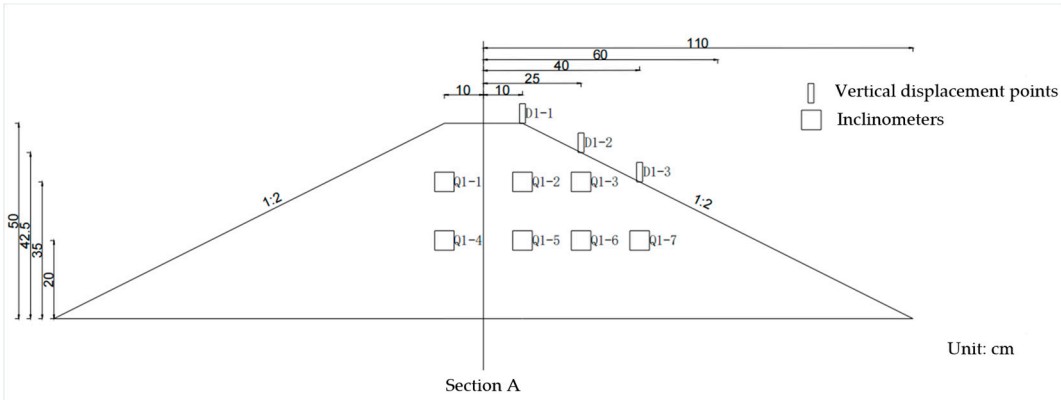

**Figure 3.** Detailed drawing of measurement point arrangement at section A.

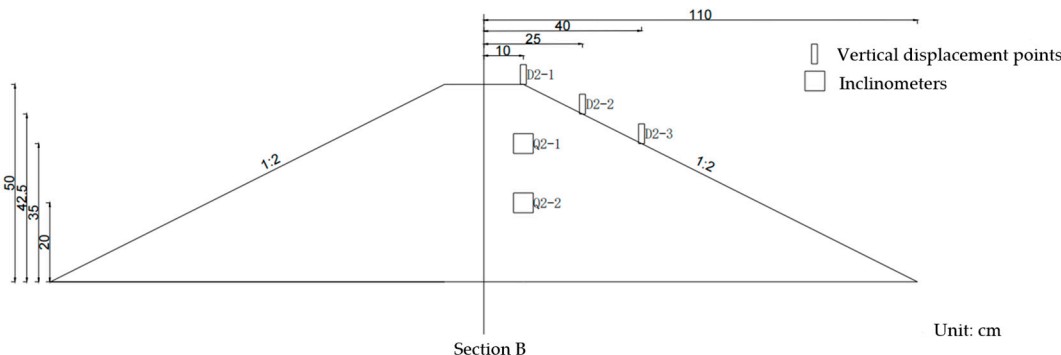

**Figure 4.** Detailed drawing of measurement point arrangement at section B.

2.4.2. Measurement Techniques

The surface deformation data of the dam are acquired by manual observation. The data auto-acquisition system monitors and collects the internal deformation data, including acquisition software, a measuring control unit (MCU), and various sensors. The acquisition frequency of the acquisition software is one time/minute. The sensors include water level meters and inclinometers, and the performance parameters of the instruments are shown in Table 2.

**Table 2.** Main performance parameters of the measuring instrument.

| Number | Measurement Items | Sensors | Model | Measurement Parameters | Measuring Accuracy | Measuring Range |
|---|---|---|---|---|---|---|
| 1 | Environmental variables | Hydrological gauge | SCYG319 | Upstream and downstream water level | ±1.02 cm | 0~102 cm |
| 2 | | Digital thermo-hygrometer | 608-H1 | Temperature and humidity | ±0.5 °C ±3%RH | 0~+50 °C 10~95%RH |
| 3 | Deformation | Level instrument | AT-B2 | Surface vertical displacements | ±15′ | 0.7 mm/km |
| 4 | | Inclinometer | CYY-QJ | Inclination angle | ±0.2° | ±45° |

The MCU collects the signals from the water level meter and inclinometer by accepting commands from the upper machine software and saves them to the upper machine. The data acquisition software realizes automatic acquisition and stores the data acquired by the MCU in the database.

### 2.5. Test Procedure

The experiment employs staged impoundment. The first impoundment started on 7 July at 18:20 and ended on 7 July at 19:23, lasting 1 h 3 min; the water level was stabilized for 97 h 39 min. The second impoundment started on 11 July at 21:02 and ended on 11 July at 21:30, lasting 28 min; the water level was stabilized for 92 h 50 min. The breaching period started on 15 July at 18:20 and ended on 15 July at 19:18, lasting 58 min. The reservoir water level rise to the dam crest lasted 48 min. Information tables for the first impoundment, the second impoundment, and the calendar time of the breach stage are detailed in Table 3 and Figure 5.

**Table 3.** Information on the duration of each stage of the overtopping failure experiment.

| Stage | Filling Stage | | First Impoundment Stage | | First Water Level Stabilization Stage | | Second Impoundment Stage | | Second Water Level Stabilization Stage | | Breach Stage | |
|---|---|---|---|---|---|---|---|---|---|---|---|---|
| | Starting Time | Ending Time | Starting Time | Ending Time | Starting Time | Ending Time | Starting Time | Ending Time | Starting Time | Ending Time | Starting Time | Ending Time |
| Time | 6-28 9:00 | 6-28 18:00 | 7-7 18:20 | 7-7 19:23 | 7-7 19:23 | 7-11 21:02 | 7-11 21:02 | 7-11 21:30 | 7-11 21:30 | 7-15 18:20 | 7-15 18:20 | 7-15 19:18 |
| Reservoir level (cm) | / | / | 0 | 26 | 26 | 26 | 26 | 42 | 42 | 42 | 42 | 37 |
| Duration | 9 h 00 min | | 1 h 03 min | | 97 h 39 min | | 0 h 28 min | | 92 h 50 min | | 0 h 58 min | |

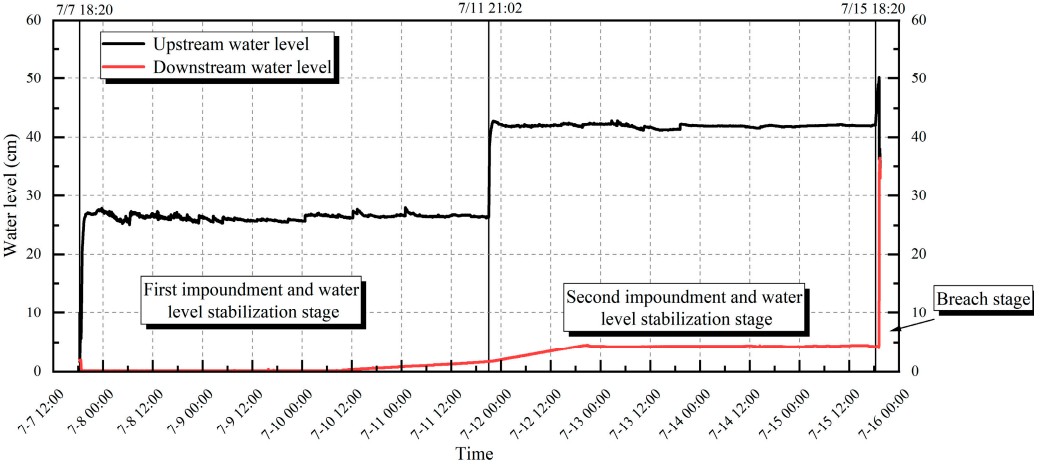

**Figure 5.** Process of upstream and downstream water level changes.

## 3. Data Analysis

The upstream and downstream water level change processes are shown in Figure 5, and the upstream and downstream water level change processes during the dam failure period are shown in Figure 6.

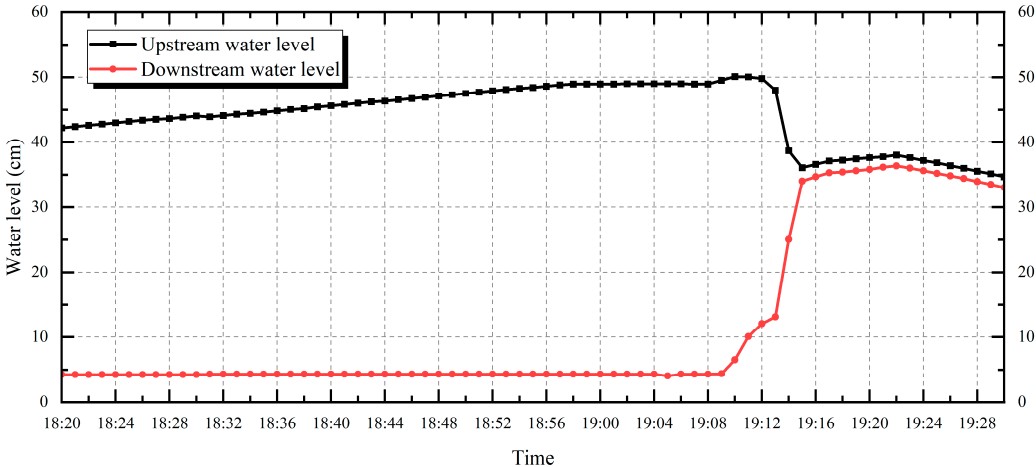

**Figure 6.** Process of upstream and downstream water level changes during the breaching stage.

The first impoundment period lasted 1 h 03 min, and the upstream water level increased from 00 cm to 26 cm; the upstream water level was maintained at 26 cm, and the length of the stabilization period was 97 h 39 min; the second impoundment period lasted 0 h 28 min, and the upstream water level increased from 26 cm to 42 cm; the upstream water level was maintained at 42 cm, and the length of the stabilization period was 92 h 50 min.

The downstream water level began to increase 63 h 27 min after the start of the first impoundment, and the downstream water level increased significantly after the start of the second impoundment and stabilized to about 4.5 cm after 120 h 14 min.

During the breaching period, the water level increased from 42 cm to the dam crest for 48 min, after which the overtopping breaching damage began. Then, the water levels upstream and downstream were leveled off after 7 min, and the breaching ended.

### 3.1. Surface Deformation

The change process of vertical displacement of the dam surface during the impoundment stage is shown in Figure 7, and the cumulative vertical displacement of the surface in each stage at each measurement point is shown in Table 4.

(1) The dam body generally showed vertical downward displacement. The cumulative vertical displacements before the breaching period in section A (the center section of the dam), i.e., the D1-1, D1-2, and D1-3 measuring points, were 5.3 mm, 5.1 mm, and 5 mm, respectively, while the cumulative vertical displacements in section B (40 cm to the right of the center section of the dam), i.e., the D2-1, D2-2, and D2-3 measuring points, were 4.9 mm, 5 mm, and 11 mm, respectively.

(2) The vertical displacement of the dam crest had the maximum amplitude of change; the closer to the downstream side of the measurement point, the smaller the amplitude of change. The measurement points with the same axis distance, the vertical displacement of section A was generally more significant than the displacement of the measurement point of section B.

(3) Nevertheless, the most considerable change in vertical displacement occurred at measurement point D2-3 in section B. According to the analysis, with the increase in the saturation line of the dam body, the soil of the downstream slope was saturated and the height of escape point increased, resulting in longitudinal cracks in the soil of the

downstream slope and slight slippage to the downstream side, which led to the considerable vertical displacement measurement value of the D2-3 measurement point.

(4) The vertical displacement of the dam body was well correlated with the impoundment process and had a certain lag. During the two impoundments and water level stabilization processes, the vertical displacement of the dam body showed a pattern of gradual increase and tended to stabilize.

(5) Combined with the whole impoundment process, the rate of change in vertical displacement during the first impoundment and water level stabilization was fast and in the rapid development stage, while the amount of change during the second impoundment and water level stabilization was not significant.

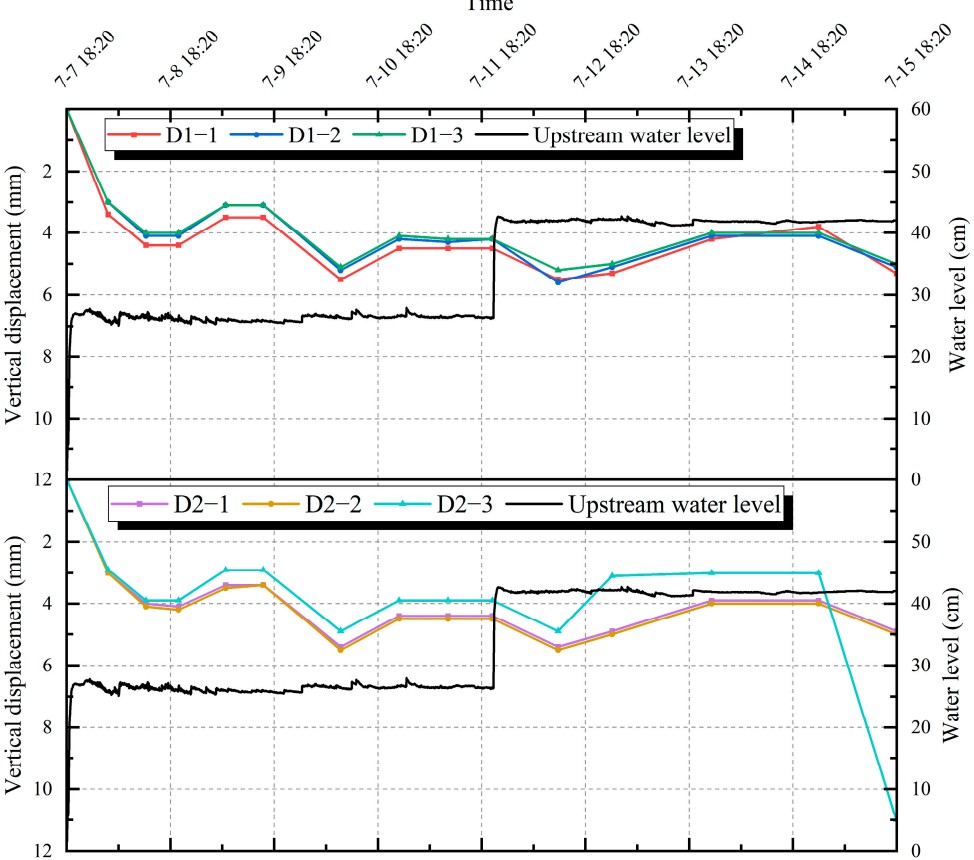

**Figure 7.** Change process of vertical displacement of the dam surface during the impoundment stage.

**Table 4.** Cumulative vertical displacement of the surface in each stage at each measurement point.

| Measuring Points | First Impoundment Stage | First Water Level Stage | Second Impoundment Stage | Second Water Level Stage |
|---|---|---|---|---|
| D1-1 | 3.4 | 4.5 | 5.5 | 5.3 |
| D1-2 | 3 | 4.2 | 5.6 | 5.1 |
| D1-3 | 3 | 4.2 | 5.2 | 5 |
| D2-1 | 3 | 4.4 | 5.4 | 4.9 |
| D2-2 | 3 | 4.5 | 5.5 | 5 |
| D2-3 | 2.9 | 3.9 | 4.9 | 11 |

## 3.2. Analysis of Internal Deformation during the Impoundment Stage

### 3.2.1. Left and Right Bank Horizontal Displacement (L&R HD)

The change process in the L&R HD at each measurement point (Q1-1, Q1-2, Q1-3, Q1-4, Q1-5, Q1-6, and Q1-7) during the impoundment stage of section A is shown in Figure 8, and the horizontal displacements of the left and right banks at each measurement point (Q2-1 and Q2-2) during the impoundment stage of section B are shown in Figure 9. The L&R HD at each measurement point in the water impoundment stage of section A shows a ladder-type jump, which is only an auxiliary reference for qualitative analysis.

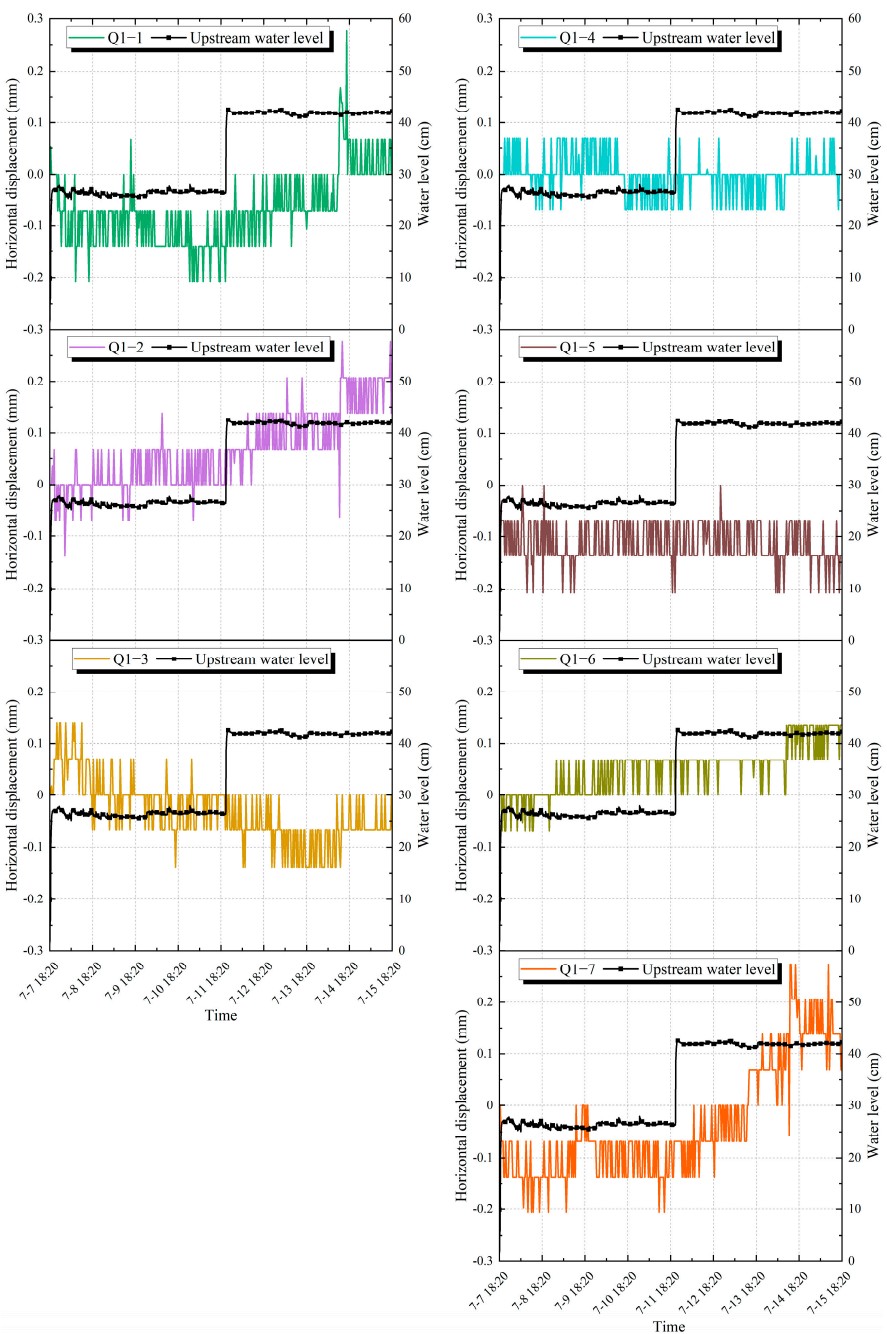

**Figure 8.** Change process in the left and right bank horizontal displacement (L&R HD) at each measurement point during the impoundment stage of section A.

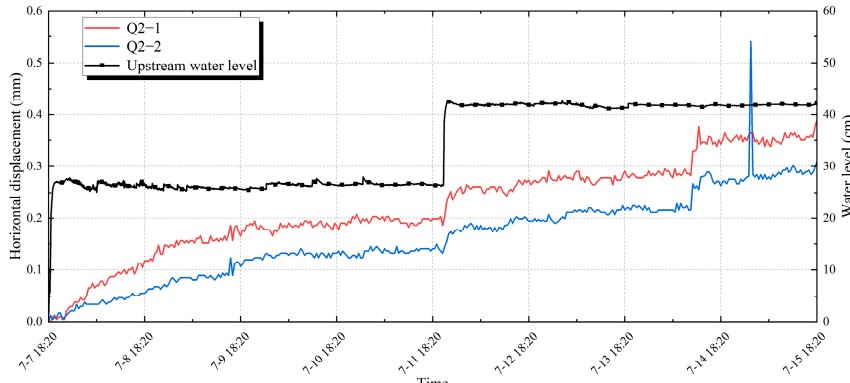

**Figure 9.** Change process in the L&R HD at each measurement point during the impoundment stage of section B.

(1) The horizontal displacement of the two sections' left and right banks shows a general deformation trend to the left bank, and the displacement of section B is more significant than that of section A. In the water impoundment and water level stabilization stage, the dam body is subject to seepage pressure and soil wetting; the deformation gradually increases, and the L&R HD tends to tilt to the left bank. As the water level of the upstream reservoir increases, the measured value of the measuring point gradually increases, and the rate of change is faster in the second impoundment period compared with the first impoundment period. Comparing the measuring points with the same elevation and the same axis distance in two different sections, such as measuring point Q2-1 with 35 cm elevation in section B and measuring point Q1-2 in section A, it is found that the measured value of the measuring point in section B has an enormous variation, which indicates that the horizontal displacement of the right bank of the dam body changes more than the location of the middle of the dam.

(2) Displacements at the measurement points near the dam crest and the dam toe are relatively more substantial. A comparative analysis of the most extensive displacement changes between the two sections in the storage stage found that the 35 cm elevation downstream side of the dam crest measurement points (Q1-2 and Q2-2) and the 20 cm elevation near the dam toe measurement point (Q1-7) measured larger value changes, so we should focus on the change in horizontal displacement on the left and right banks in this area.

(3) The higher the elevation and the larger the axis distance of section A, the relatively more significant the L&R HD is. The upstream and downstream displacements near the dam crest show almost the same trend (e.g., measurement points Q1-1 and Q1-2 at 35 cm elevation). For the measuring points with the same axis distance, the increase in measured value of the points with higher elevation is greater than that of the points with lower elevation (e.g., measurement points Q1-2 at 35 cm elevation and Q1-5 at 20 cm elevation, as well as Q1-3 at 35 cm elevation and Q1-6 at 20 cm elevation), which means that the higher the elevation of the points, the higher the influence of the dam settlement and seepage pressure, resulting in more remarkable changes in the left and right banks' horizontal displacements. For the measuring points with the same elevation, the larger the axis distance, i.e., the measurement points far away from the dam crest, the more significant the change in the measured value is (e.g., measurement points Q1-4, Q1-5, Q1-6, and Q1-7 at 20 cm elevation), which is inferred to be due to the wetting of the downstream dam material because of the lack of drainage facilities in the dam body.

(4) For the measurement points in section B, the measured values of the two measurement points show almost the same elevation pattern, and there are three rapid elevation phases, respectively. After the first impoundment, by the action of seepage pressure, the measured values of Q2-1 (35 cm elevation) and Q2-2 (20 cm elevation) began to

increase rapidly and slowed down after the first impoundment for about 48 h. After the second impoundment, the measured values of the two points began to increase rapidly and gradually slowed down after the first impoundment for about 1 h 10 min. With the upstream reservoir water level increasing, the seepage pressure of the dam body increases. A minor slippage was observed on the downstream slope of the dam, and the soil in the dam body was redistributed by force, which led to a third rapid increase in the measured values and the gradual formation of a stable phase of elevation after the rise. The same pattern is shown across section A measurement points at different elevations but with the same axis distance.

### 3.2.2. Upstream and Downstream Horizontal Displacement (U&D HD)

The change process in U&D HD at each measurement point (Q1-1, Q1-2, Q1-3, Q1-4, Q1-5, Q1-6, and Q1-7) during the impoundment stage of section A is shown in Figure 10, and the changes in U&D HD at each measurement point (Q2-1 and Q2-2) during the impoundment stage of section B are shown in Figure 11. The U&D HD at each measurement point in the water impoundment stage of section A shows a ladder-type jump, which is only an auxiliary reference for qualitative analysis.

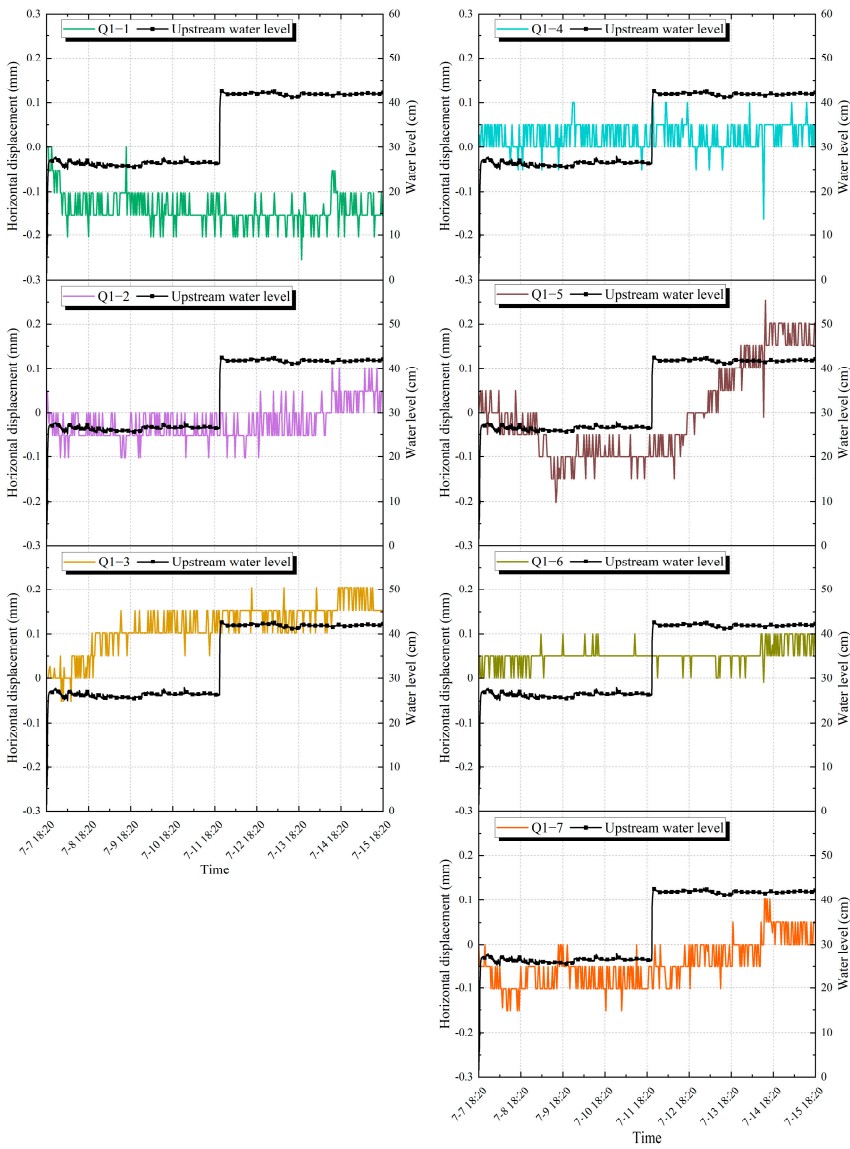

**Figure 10.** Change process in the upstream and downstream horizontal displacements (U&D HD) at each measurement point during the impoundment stage of section A.

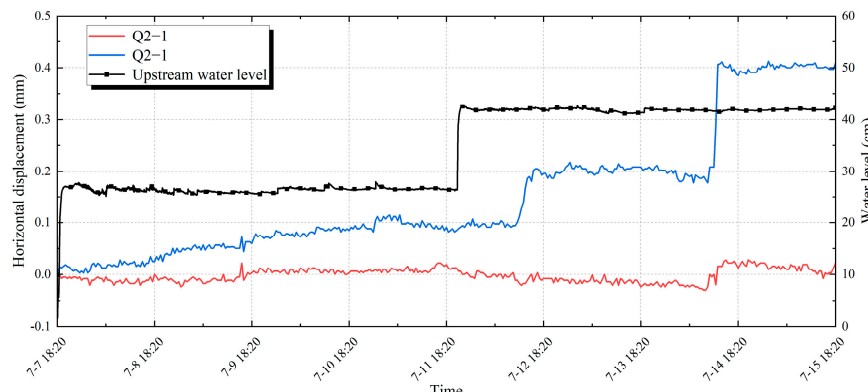

**Figure 11.** Change process in the U&D HD at each measurement point during the impoundment stage of section B.

(1) The magnitude of change in U&D HD is more significant in the later stages than in the earlier stages. The deformation of the dam body is gradually increased by seepage pressure and soil wetting during impoundment and water level stabilization, and the U&D HD generally shows an inclination to the downstream side. Compared with the first impoundment and water level stabilization period, the displacement change rate is faster in the second impoundment and water level stabilization period.

(2) The displacement of the right dam section to the downstream side is greater than that of the middle section, and the displacement of the measurement points near the downstream dam slope location is relatively significant. While comparing the measuring points with the same elevation and axis distance in two different sections, it is found that the variation in the measured value of the measuring point of section B is more prominent (e.g., measurement points Q2-2 at 20 cm elevation in section B and Q1-5 at 20 cm elevation in section A), which indicates that the variation in U&D HD in the right dam section is more considerable than that of the middle position of the dam. In the comparative analysis of the displacement of the water impoundment stage at the most prominent points of the two sections, we found that the measurement points (Q1-5 and Q2-1) at 20 cm elevation downstream of the dam crest measured the most significant variation, so we should focus on the changes on U&D HD on the downstream side of the dam crest.

(3) For each measurement point in section A, the U&D HD exhibits an overall trend of tilting upstream first. With the gradual stabilization of the saturation line of the dam body, upstream tilting changed to a downstream tilting trend. For measuring points at the same elevation, the smaller the axis distance is, i.e., the closer the measuring point is to the dam crest, the more significant the change in the measured value is (e.g., measurement points Q1-5 and Q1-7 at 20 cm elevation). For measuring points with the same axis distance, the lower the elevation, the larger the amplitude of change in the measured value (e.g., measurement points Q1-2 at 35 cm elevation and Q1-5 at 20 cm elevation).

(4) For the measurement points in section B, the whole change rule of the two measurement points is the same; the overall change amplitude at the 35 cm elevation measurement point Q2-1 is smaller than the 20 cm elevation measurement point Q2-2, which indicates that, for measurement points at the same axis distance but different elevation, the lower the elevation, the more extensive the amplitude of the value of the change is, which is the same rule as the one shown in the measurements of section A. The measured value of point Q2-2 starts to increase 6 h 27 min after the first impoundment, reaches its peak value after 71 h 30 min, and then gradually stabilizes; it starts to increase rapidly after 12 h 50 min from the second impoundment, reaches its peak value after 6 h 30 min, and then gradually stabilizes. With the slight slippage on the downstream dam slope, the measured values of the two points appear to

increase rapidly for a short period of time and then gradually stabilize. The measured value of the 20 cm elevation Q2-2 point in section B has a higher correlation with the upstream reservoir level, with some lag. Relative to the first impoundment and water level stabilization period, the rate of displacement change is faster during the second impoundment and water level stabilization period.

### 3.3. Analysis of Internal Deformation during the Breach Stage

It should be especially pointed out that, due to the lack of seepage control measures for the dam body, a certain degree of infiltration damage occurred before the overtopping breach, which resulted in the collapse of the right bank of the dam body during the overtopping breach without the collapse of the upstream side of the center section of the original test design breach.

The evolution of the dam failure during the breaching period is shown in Figure 12.

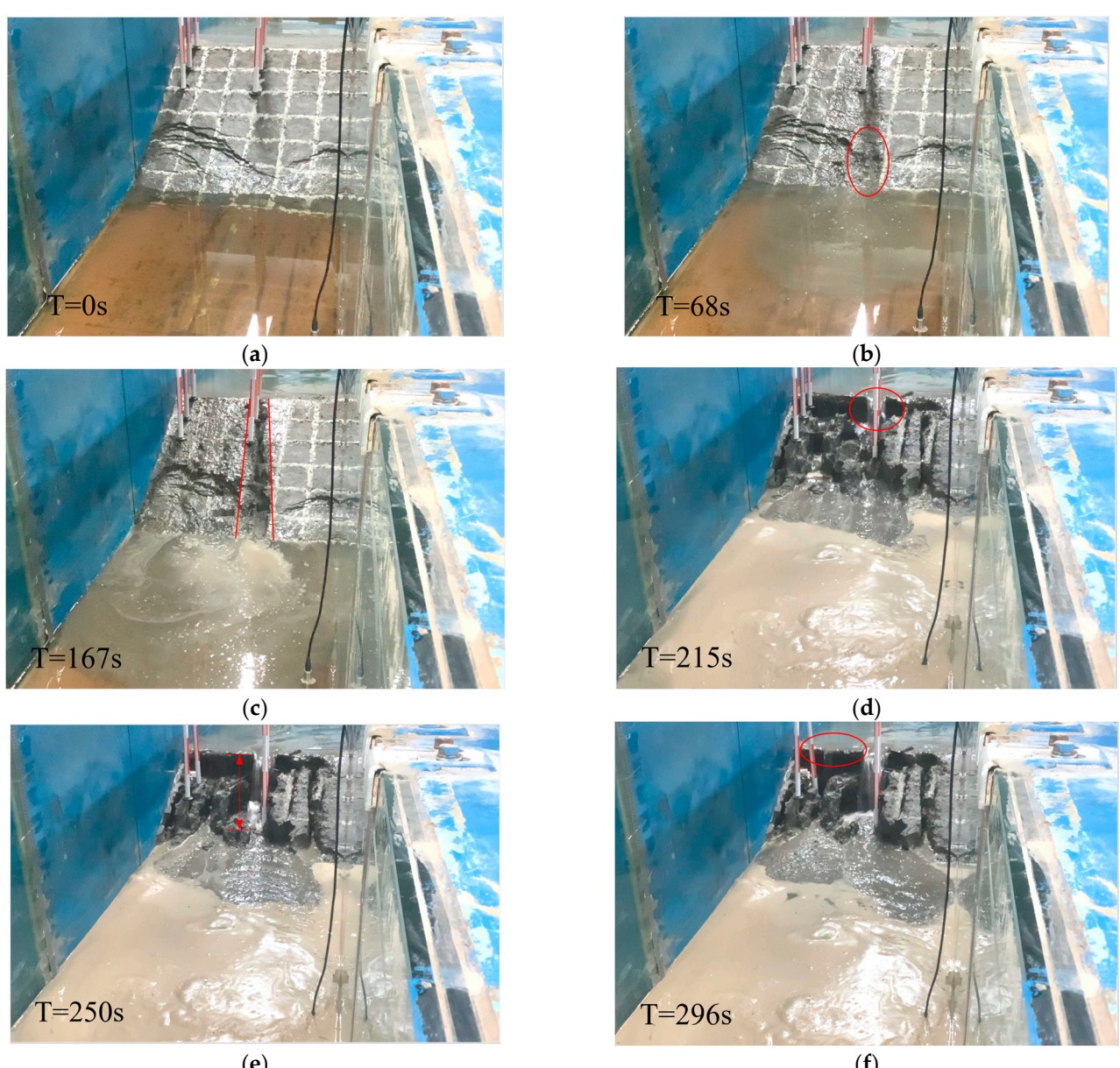

**Figure 12.** *Cont.*

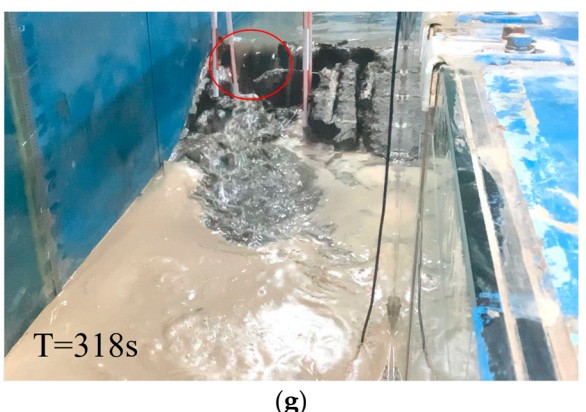

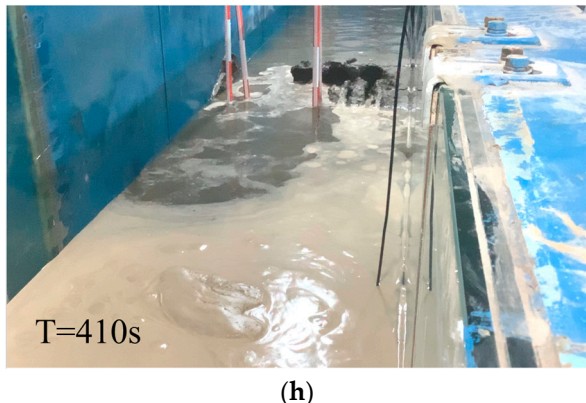

(**g**)

(**h**)

**Figure 12.** Evolution of the dam breaching process: (**a**) the upstream reservoir overtopped; (**b**) "head-cut" erosion began to form and developed from the downstream to the upstream side; (**c**) formation of multilevel headcut erosion; (**d**) the "headcuts" merged to form a "waterfall-like water flow"; (**e**) the water flow drop increased and the scouring capability further increased; (**f**) the soil on the upstream side of the dam crest in the right dam section was destabilized and collapsed due to gravity; (**g**) the breach continued expanding and gradually formed structural collapses; (**h**) the upstream and downstream water levels leveled off and the failure ended.

### 3.3.1. Left and Right Bank Horizontal Displacement (L&R HD)

The variation process of internal L&R HD at each measurement point is shown in Figure 13, and the statistical situation of the variation in the measured values is shown in Table 5.

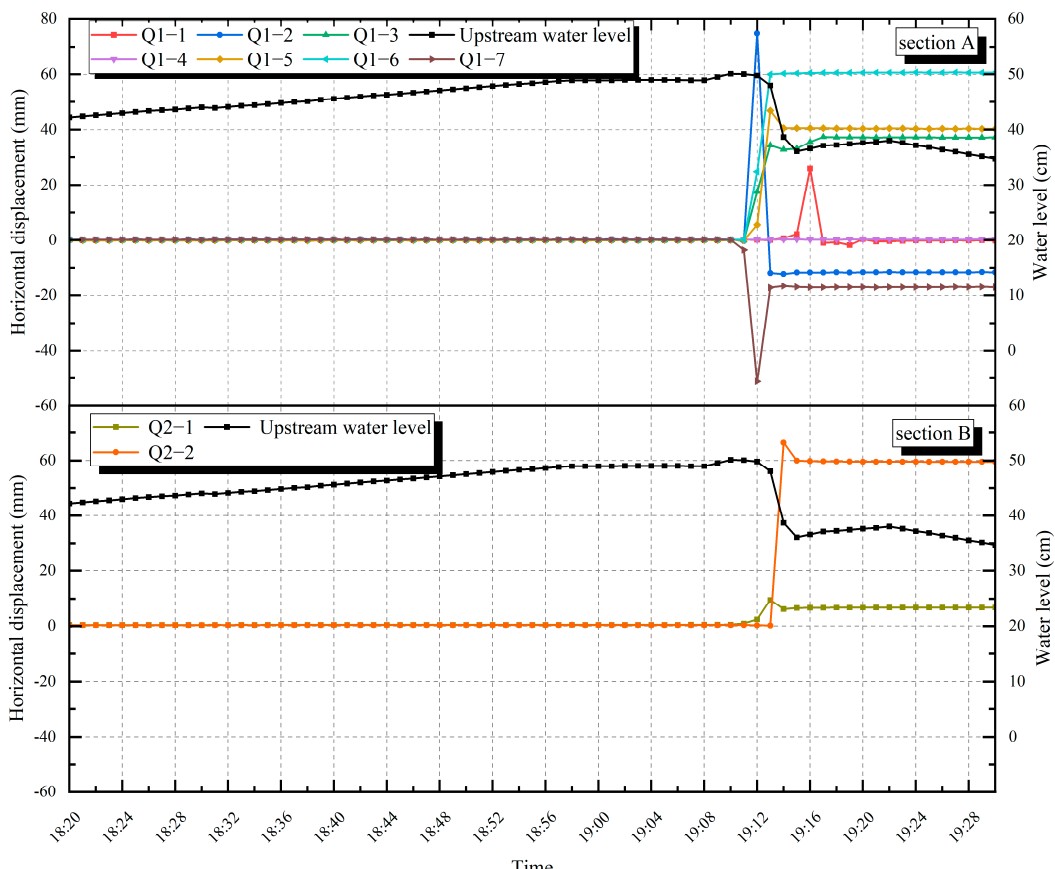

**Figure 13.** Change process of the L&R HD at each measurement point during the breach stage of sections A and B.

**Table 5.** Time and quantity of the sudden change in the L&R HD at the measurement points of sections A and B.

| Measuring Points | Time of Sudden Change (hh:mm) | Before the Sudden Change (mm) | After the Sudden Change (mm) | Amount of Sudden Change (mm) |
|---|---|---|---|---|
| Q1-1 | 19:12 | 2.29 | 25.97 | 23.68 |
| Q1-2 | 19:12 | −0.35 | 74.85 | 75.20 |
| Q1-3 | 19:12 | −0.07 | 17.54 | 17.61 |
| Q1-4 | | No sudden change | | |
| Q1-5 | 19:13 | 5.67 | 46.86 | 41.19 |
| Q1-6 | 19:12 | 0.21 | 24.84 | 24.63 |
| Q1-7 | 19:12 | −3.58 | −51.21 | −47.63 |
| Q2-1 | 19:13 | 2.41 | 9.53 | 7.12 |
| Q2-2 | 19:14 | 0.19 | 66.45 | 66.26 |

During the process of the upstream water level rising to the dam crest, the variation rule of each measuring point is the same as that of the impoundment stage, and the L&R HD of the dam body still shows the trend of tilting to the left bank in general. As the reservoir water level increases further, the dam body saturation line continues to elevate, and the height of escape point of downstream slope increase, resulting in further expansion of the downstream dam body slip. The same situation occurs on the left bank of the dam body. The breaching process is mainly divided into the following stages:

(1) The reservoir water overtopped the dam at 19:08:30. It began to erode the downstream slope of the dam; the erosion process was slow and gradual, and the upstream reservoir water level was still in the stage of slow elevation. With the gradual increase in overtopping erosion capacity, "multilevel headcut" erosion was formed after 68 s and gradually developed from the dam toe to the upstream side of the dam crest. The measurement point Q1-7 on the downstream side of the 20 cm elevation at the center section of the breach (section A) first started to change slowly to the right bank at 19:10.

(2) With the "multilevel headcut" continuing to erode back toward the dam and developing upstream, the "multilevel headcut" gradually merged into a larger "headcut". Within 215 s after the breach formed, a "waterfall-like water flow" emerged at this point, signifying the dam body erosion damage had progressed to the dam crest. At the center section of the breach (the downstream side of the dam crest and downstream side measurement points Q1-2 and Q1-3 at 35 cm elevation and the downstream side measurement points Q1-6 and Q1-7 at 20 cm elevation), the measurement values for above measurement point suddenly changed at 19:12.

(3) Subsequently, the erosion capacity of the "waterfall-like water flow" became more vigorous. The water flow continuously eroded the dam body at the water tongue fall point. The water flow drop increased, and the scouring capability further increased while continuously eroding the bottom. The measured values of Q1-5 at the downstream side of the dam crest at 20 cm elevation in the center section of the breach and Q2-1 at the downstream side of the dam crest at 35 cm elevation in section B suddenly changed at 19:13.

(4) With the rapid erosion of the dam body by the "waterfall-like water flow", after 296 s of upstream reservoir water overflowing over the dam crest, the right bank of the dam crest soil body collapsed due to gravitational instability, resulting in the formation of a new breach whose width increased rapidly. At this moment, the failure and collapse processes of the dam body were very intense, the water erosion ability was potent, the flow and flow velocity of the breach increased rapidly, and the water level of the upstream reservoir decreased rapidly. The dam body formed a structural collapse after 410 s, resulting in the upstream and downstream reservoirs being connected. The measurement value of Q2-2 at the downstream side of the dam crest at 20 cm

elevation in section B suddenly changed at 19:14, and the measurement value of Q1-1 at the upstream side of the dam crest at 35 cm elevation at the center section of the breach suddenly changed at 19:16.

(5)  As shown in Table 5, the sudden change in measuring point Q1-2 on the downstream side of the dam crest at 35 cm elevation in section A was the most considerable, with an amplitude of 79.2 mm. Before the structural collapse of the dam body occurred, embankment dam failure mainly occurred on the downstream side of the dam body. The dam crest's downstream side and the dam body's downstream side in section A have considerable overall variation. When combined with the breaching process, it can be seen that the initial breaching mainly occurred in section A. Compared with the same location of measuring points in section A, the sudden change time of the measuring points in section B took place later, and the amount of sudden change was smaller (e.g., measuring point Q2-1 at the downstream side of the dam crest at the 35 cm elevation). Due to the formation of a new breach on the right bank of the dam body, the amount of sudden change on the downstream side of the dam crest measuring point Q2-2 at 20 cm elevation in section B was 66.26 mm, which is more extensive than that of the original breach center section at the same location. From the beginning to the end of the overtopping breach of the earth dam, there was no sudden change in the measurement point on the upstream side of the dam crest at the 20 cm elevation in section A.

### 3.3.2. Upstream and Downstream Horizontal Displacement (U&D HD)

The variation process of internal U&D HD at each measurement point is shown in Figure 14, and the statistical situation of the variation in the measured values is shown in Table 6.

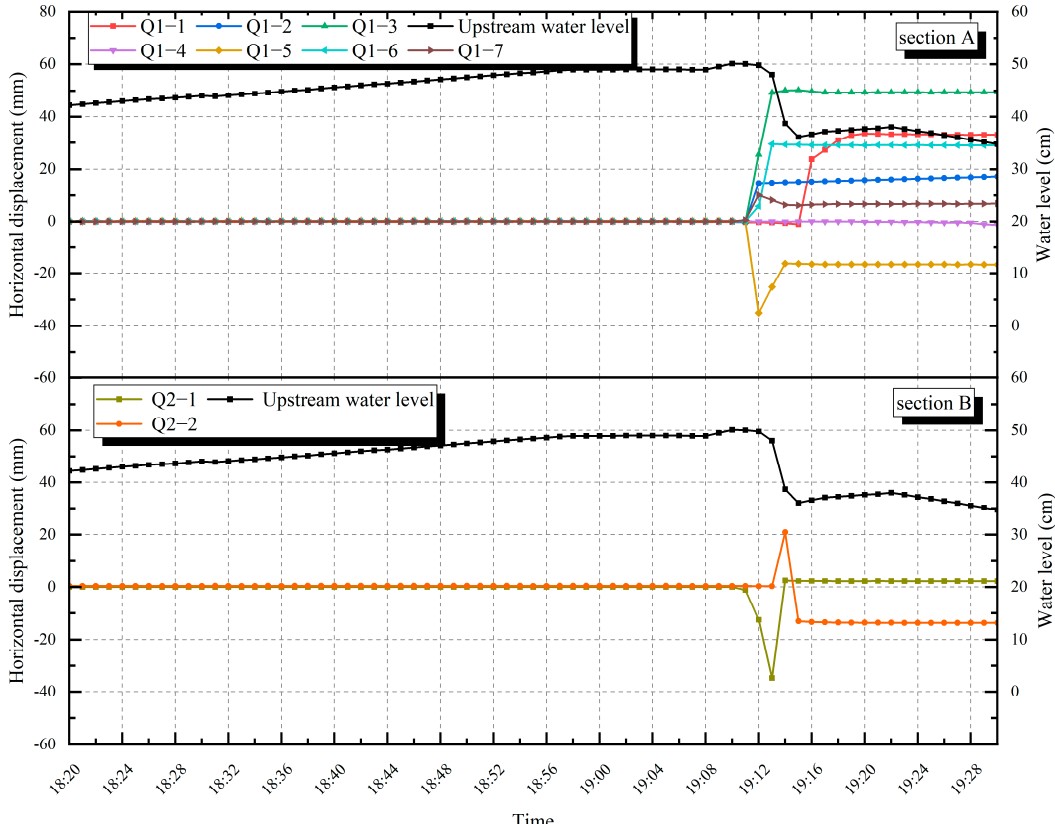

**Figure 14.** Change process of the U&D HD at each measurement point during the breach stage of sections A and B.

**Table 6.** Time and quantity of the sudden change in the U&D HD at the measurement points of sections A and B.

| Measuring Points | Time of Sudden Change (hh:mm) | Before the Sudden Change (mm) | After the Sudden Change (mm) | Amount of Sudden Change (mm) |
|---|---|---|---|---|
| Q1-1 | 19:16 | −1.12 | 23.77 | 24.89 |
| Q1-2 | 19:12 | −0.26 | 14.72 | 14.98 |
| Q1-3 | 19:12 | −0.10 | 25.45 | 25.55 |
| Q1-4 | | No sudden change | | |
| Q1-5 | 19:12 | 0.15 | −34.98 | −35.13 |
| Q1-6 | 19:12 | 0.05 | 5.79 | 5.74 |
| Q1-7 | 19:12 | 0.51 | 10.07 | 9.56 |
| Q2-1 | 19:12 | −1.15 | −12.52 | −11.37 |
| Q2-2 | 19:14 | 0.31 | 20.92 | 20.61 |

During the upstream water level increase to the dam crest, the rule of change in each measurement point is the same as that of the impoundment stage, and the U&D HD of the dam body still presents a downstream tilt trend in general. Combined with the four stages of the breaching process described in the previous section, the time sequence law of each measurement point is the same as the time law of the sudden change in the left and right bank displacement measurements, which is not repeated here.

As can be seen from Table 6, the downstream side measuring point Q1-5 of the dam crest at 20 cm elevation in section A has an immense amount of mutation, with a variation of 35.13 mm. Compared with the upstream side of the dam crest and the downstream side of the dam body, the U&D HD mutations of the measuring points on the downstream side of the dam crest are extensive, such as those of downstream side measuring point Q1-5 of the dam crest at 20 cm elevation in section A and downstream side measuring point Q2-2 of the dam crest at 20 cm elevation in section B. There is no sudden change in the measurement point on the upstream side of the dam crest at the 20 cm elevation in section A, consistent with the horizontal displacement changes of the left and right banks.

3.3.3. Warning Signs of Deformation Analysis

(1) The overall direction of the dam overtopping breach sudden change is toward the left bank and downstream, consistent with the changing trend of the water impoundment stage.

(2) After the overtopping failure started, the closer to the downstream slope of the dam, the earlier the sudden change appeared, such as the 20 cm elevation Q1-7 measuring point in section A.

(3) The higher the measuring point, the earlier the sudden change time, such as the 35 cm elevation Q1-2 and 20 cm elevation Q1-5 measuring points in section A.

(4) That is, the time sequence of the sudden change in the measured value of each measurement point is from the downstream side to the upstream side and from the higher position to the lower position of the dam elevation, which is in accordance with the fundamental process of overtopping and breaching of embankment dams [30–32].

(5) As the erosion capacity of water flow in section A (the center section of the breach) is more significant than that in section B, the time of sudden change in the measurement point in section B is slightly later than that in the center section of the breach.

(6) Before the dam body's structural collapse occurred, the measurement point on the upstream side had not caused any sudden changes.

In case the embankment dam encounters overtopping danger, it can be combined with the internal deformation field changes of the dam body to make a comprehensive study of the security state of the dam body and provide early warnings for the safety of the dam body.

## 4. Conclusions

In this study, real-time monitoring data of the internal deformation field of a model dam during the whole process of impoundment, operation, and overtopping were obtained by carrying out an indoor small-scale model experiment simulating the overtopping of homogeneous embankment dams, embedding inclinometers inside the earth dam model, and setting vertical displacement measurement markers on the surface. Through qualitative and quantitative analyses of the measured data, the following conclusions were drawn:

(1) During the impoundment stage, with the elevation of the water level, the overall horizontal displacement inside the dam changed in the direction of the left bank and downstream side. The rate of displacement deformation was more rapid in the second impoundment and water level stabilization period compared to the first impoundment and water level stabilization period. The deformation was strongly correlated with the upstream reservoir level, with a certain lag. The internal horizontal displacement and left and right bank displacement variations at the downstream side of the dam crest were generally large, so we should focus on this area.

(2) The failure process can be divided into four stages: the downstream dam slope erosion gully formation and development stage; the downstream dam slope "multilevel headcut" erosion stage; the breach "waterfall-like water flow" erosion stage; and the upstream side of the dam body rapid collapse stage. The time sequence of the sudden change in the horizontal displacement at each measurement point is from the downstream side to the upstream side and from the higher elevation of the dam body to the lower elevation, which is consistent with the basic process of overtopping and breaching of embankment dams. The downstream side of the dam crest and the downstream side of the dam body within the measurement point of the left and right banks of the horizontal displacement of the overall sudden change are large, and the downstream and upstream horizontal displacement within the measurement point of the downstream side of the dam crest has a large amount of sudden change. Before the dam body suffered structural collapse, no sudden change had occurred at the measuring point on the upstream side of the dam center section.

(3) Due to the lack of seepage control facilities at the dam, during the stabilization process of the upstream reservoir water level at 42 cm, longitudinal cracks emerged and the height of escape point increased within the 0~0.3 m elevation range of the dam slope surface on the right dam section. As the upstream reservoir water level increased, the cracks slightly increased further to the downstream measurement of the slip, resulting in a decrease in soil compactness on the right side. At the stage of impoundment, the internal horizontal displacement of the right bank of the dam body was larger than that of the middle of the dam body. Meanwhile, the right bank of the dam body was in contact with the rigid boundary of the flume, which was a weak contact zone, and the right bank of the dam body experienced structural collapse during the final overtopping failure.

(4) When an embankment dam encounters overtopping risks, attention should be focused on the thinner sections of the dam body, such as the contact point between the dam body and the two sides of the mountain, as well as locations with higher seepage pressure. This should be combined with real-time variation in monitoring data so that immediate emergency treatment measures can be taken.

Although this study has achieved some results, there are some limitations: First of all, while the model was overtopping, no sudden change occurred in the measured value of the upstream side of the dam crest at the 20 cm elevation in section A. Since the experimental design did not consider installing inclinometers on the upstream side of the dam crest in section B, it was not possible to compare and judge the sudden change in the measured values on the upstream side of the dam crest. Secondly, the test instrumentation was mainly placed in the breach's center section, and mainly the section's deformation field at the instrumentation location was obtained. Since no numerical simulation was performed in this study, the deformation field inside the whole model dam body could not be obtained.

Another limitation of this study is that the flume was small and the water–soil coupling process was sophisticated, making it difficult to mirror the original model. Further in-depth research will be carried out in order to provide support for the theory of embankment dam overtopping warnings.

**Author Contributions:** Conceptualization, Y.G. and S.W.; Data curation, Q.L.; Formal analysis, Q.L.; Funding acquisition, Y.G. and S.W.; Investigation, Q.L., X.L. and H.W.; Methodology, Q.L.; Project administration, Y.G.; Resources, Q.L., X.L. and H.W.; Software, X.L. and H.W.; Supervision, Y.G. and S.W.; Validation, Q.L., Y.G. and S.W.; Visualization, Q.L.; Writing—original draft, Q.L.; Writing—review and editing, Q.L., Y.G. and S.W. All authors have read and agreed to the published version of the manuscript.

**Funding:** This research was funded by National Natural Science Foundation of China (51979175), Nanjing Hydraulic Research Institute's Basic Scientific Research Business Fund Scientific Research and Innovation Team Building Project (Y722003), Nanjing Hydraulic Research Institute's Basic Scientific Research Business Fund Scientific Research and Innovation Team Building Project (Y520009-1) and Nanjing Hydrological Research Institute's Basic Scientific Research Business Fee Leading Project (Y723008).

**Data Availability Statement:** The raw/processed data of the findings of this study cannot be shared at this time because they are part of an ongoing study. Corresponding authors may be contacted if necessary.

**Acknowledgments:** We sincerely thank all the reviewers and editors for their professional comments and suggestions regarding this manuscript.

**Conflicts of Interest:** The authors declare no conflict of interest.

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
