# Peer review of "Deformation Field Analysis of Small-Scale Model Experiment on Overtopping Failure of Embankment Dams"

_water, doi:10.3390/w15244309_

Round 1
Reviewer 1 Report
Comments and Suggestions for Authors
In this experiment, downstream water will be stored, but in reality, downstream water will flow directly away. How about its influences on the failure mode.
Comments on the Quality of English LanguageThe two horizontal displacement "horizontal displacement of the left and right banks“ and "upstream and downstream horizontal displacement" is ambiguous, the definition of direction may be abbreviated.
Reviewer 2 Report
Comments and Suggestions for Authors
1、Line 15, further streamlines some of the conclusions drawn from this study in the Abstract section.
2、Line 134, complements the location of the video surveillance points in the Figure a.
3、Line 137, the mechanical parameters of the soils used in the experiment are suggested to be described in a table and add median grain size, soil code, and classification.
4、Line 168, completes the specific positions of the benchmarks in Figure c.
5、Line 215, removes the internal vertical box lines from the table.
Comments on the Quality of English LanguageMinor editing of English language required.
Reviewer 3 Report
Comments and Suggestions for Authors
This work analyzed the deformation field characteristics of embankment dams during the overtopping failure with the small-scale model test. The topic is interesting and experimental results revealed the surface deformation and internal deformation in the impoundment stage and breach stage. The structure of manuscript is well organized, while there are some issues that should be revised:
(1) “Symbol: cm”in Figure 1->"Unit: cm".
(2) It might be better to show the location of initial breach on the model dam in Figure 2 for a batter description.
(3) Figures 3 and 4 are not clear enough.
(4) Line 252: "0+0.5 section" should be "0+0.9 section".
(5) Figure 11: two lines are for Q2-1.
(6) There might be too many discussions in “Data Analysis” and this part could be more concise.
Reviewer 4 Report
Comments and Suggestions for Authors
Dear Editor
In this work, the authors construct a small-scale, test model of a standard homogeneous embankment dam to study the failure due to overtopping. Through a set of measurements installed in the model -displacement sensors on the surface of the dam and inclinometers within the breach- the effects of the whole operation process, impoundment, normal operation, overtopping and failure, are logged. Authors state that these records provide a theoretical basis for the prediction and early warming of the overtopping failure.
I believe that this is a laborious and meticulous work that deserves to be known by the scientific community and, in that sense, I would recommend its publication in this journal. Such a detailed study of the dam's behavior certainly helps to establish 'project criteria' that potentially could be applied to real models. In this sense, the signs of deterioration listed in section 3.3.3, derived from the analysis of the monitored deformations, are or can be a guide 'for a comprehensive study of the security state and provide early warming for the safety of the dam body'. From my point of view, this is the argument that most justifies the dissemination of this work.
But some information and explanation should be given to understand the limitations of the model, so the results can be useful. I recommend not to publish the paper in the present form. MAYOR REVISION should be carried out.
Main comments:
1: Theoretical context:
I must say something that I consider very important about what we know as 'physical modelling'. An issue that the authors dispatch by simply saying the following (p.2, lines 63-64): ‘Physical modelling od dam failure is an essential means of reproducing the process of dam failure and revealing such mechanism´. And, later (lines 66-67): “The problems of stress and flow dissimilarity in a small-scale model have been solved to a certain extent”. A succinct sentence without any references that says rather little. Finally, in the last paragraph of the section Conclusions (p.20, lines 580-584), the authors return to the issue: 'Another limitation of the study is that the flume is small and the water-soil coupling processes is sophisticated, which cannot accommodate the similarity with the original model
I miss, either in the introduction or as a separate section, an exposition and discussion of how the 'Principle of Similitude' (see, for example, J.C Gibbings, Dimensional Analysis. Pringer 2011) has been applied to this small-scale model. How the similarity problems have been solved? To what extent are the results of the proposed model applicable to large dams?
It is far from being a question of estimating the collapse times of real dams from the times obtained in this model (section 3.3.3). We are not yet ready for this purpose. But we do need to mention something about the principle of similarity, a brief description of the difficulties in complying with this principle and some references to enlighten the reader on this important subject in relation to the small-scale model studied. We know that geometric similarity is not enough to create reliable models. Instead, it is the independent dimensionless groups, generally difficult to obtain in such complex problems, that determine the fulfillment of the above-mentioned principle.
Can the authors improve the theoretical content of this paper with a section or at least a few paragraphs informing the reader of the similarity issues of the proposed model (not strictly on a geometric scale)? I sincerely believe that this would improve the present manuscript.
2: Materials and model
Line 139: Sands are no cohesive materials unless they are cemented. Which was the way to obtain the value of cohesion? Is there any shear test results?
The use of permeable non-cohesive materials leads to seepage, heave and piping. I can be easily deduced from Figure 5 and Figure 12 (T=0), apart from horizontal displacement in downstream side of the bank. So, this experiment is not only dealing with overtopping, but with piping and heave caused by seepage forces. It should be specifically mentioned in the paper.
Data related to sands should be included: grain size and any coefficient related to Sphericity. The rockfill dam is a type of embankment dam. The scale model should inform about the relation “grainsize VS dam height”. Here, it was (I suppose) ranging from 0.02/50 to 0.08/50. Does it fit with real rockfill dams? Please, show an example.
About the model, impervious curtain is not included in the upstream side. How can a permanent granular dam storage the water?
3: Conclusions
The conclusion should be rewritten accordingly to these comments. The setting of “the prediction and early warming” should be highlighted and explained. If it´s not the case, It should be deleted from objectives
Minor comments:
1. Lines 9-10: avoid redundance.
2. Lines 18-19. If this is one dimensional strain, it should be mentioned.
3. Line 67: extent is repeated.
4. Line 83: please, rewrite it.
5. Line 118: what do you mean with the word excavated?
6. Line 144 The relative density in standards or normative is as follow:
It does not seem clear in the manuscript the usefulness of the expression (1) relative to the relative density. I think its usefulness should be more explicitly clarified. Please, justify your formula.
7. (Section 2.1) The nomenclature of the scale model sections in the drawing and in the text do not seem to match accurately.
8. I don't quite understand drawing (d) in figure 1.
9. The nomenclature to indicate the section of the breach and the piles is rare: 0+0.5 section, 0+0.9 section, pile no. 0+0.5, pile no. 0+0.9... (text of figures 3 and 4 and text of section 2.3.1).
10. Section 2.3.2 . What is MCU?
11. What the authors call 'permeability coefficient' is what is more generally referred to as 'hydraulic conductivity' (m/s)? Please, modify it.
12. The ‘Characterization of dam failure’ is an information duplicated: (p.17, section 3.3.1, and p. 19, section 3,3,2.
13. The work should be reviewed by an expert in the language. It is common to find strings of nouns and adjectives in the text that are too long and should be separated by propositions. E.g.: i) p. 3 ‘indoor small scale homogeneous embankment dam overtopping failure model’.
Comments on the Quality of English LanguageThe work should be reviewed by an expert in the language. It is common to find strings of nouns and adjectives in the text that are too long and should be separated by propositions. E.g.: i) p. 3 ‘indoor small scale homogeneous embankment dam overtopping failure model’.
Reviewer 5 Report
Comments and Suggestions for Authors
This is an experimental work that did not include a numerical model. Therefore, the authors must explain this work limitation as well as its consequence for the work relevance.
Round 2
Reviewer 3 Report
Comments and Suggestions for Authors
The manuscript has been revised according to the comments. Only one comment: What is the Figure 1(d)? Only 3 cm, 4 cm, 10 cm and initial breach can be seen in Figure 1(d).
Reviewer 4 Report
Comments and Suggestions for Authors
Dear Editor.
Authors have adressed the comments. I´m satisfied with their work and the answers to my review, so I consider the manuscript is ready for publication.
Author Response
Thank you for your encouraging comments on our work.
